# Multi-Resin Masked Stereolithography (MSLA) 3D Printing for Rapid and Inexpensive Prototyping of Microfluidic Chips with Integrated Functional Components

**DOI:** 10.3390/bios12080652

**Published:** 2022-08-17

**Authors:** Isteaque Ahmed, Katherine Sullivan, Aashish Priye

**Affiliations:** Department of Chemical and Environmental Engineering, University of Cincinnati, Cincinnati, OH 45221, USA

**Keywords:** 3D printing, microfluidics, stereolithography, microfluidic valve, microfluidic pump, resin

## Abstract

Stereolithography based 3D printing of microfluidics for prototyping has gained a lot of attention due to several advantages such as fast production, cost-effectiveness, and versatility over traditional photolithography-based microfabrication techniques. However, existing consumer focused SLA 3D printers struggle to fabricate functional microfluidic devices due to several challenges associated with micron-scale 3D printing. Here, we explore the origins and mechanism of the associated failure modes followed by presenting guidelines to overcome these challenges. The prescribed method works completely with existing consumer class inexpensive SLA printers without any modifications to reliably print PDMS cast microfluidic channels with channel sizes as low as ~75 μm and embedded channels with channel sizes as low ~200 μm. We developed a custom multi-resin formulation by incorporating Polyethylene glycol diacrylate (PEGDA) and Ethylene glycol polyether acrylate (EGPEA) as the monomer units to achieve micron sized printed features with tunable mechanical and optical properties. By incorporating multiple resins with different mechanical properties, we were able to achieve spatial control over the stiffness of the cured resin enabling us to incorporate both flexible and rigid components within a single 3D printed microfluidic chip. We demonstrate the utility of this technique by 3D printing an integrated pressure-actuated pneumatic valve (with flexible cured resin) in an otherwise rigid and clear microfluidic device that can be fabricated in a one-step process from a single CAD file. We also demonstrate the utility of this technique by integrating a fully functional finger-actuated microfluidic pump. The versatility and accessibility of the demonstrated fabrication method have the potential to reduce our reliance on expensive and time-consuming photolithographic techniques for microfluidic chip fabrication and thus drastically lowering our barrier to entry in microfluidics research.

## 1. Introduction

Microfluidics is the study and manipulation of fluids at microscopic scales (usually sub-millimeters), where micro-scale physics dominate. In recent years, microfluidics has quickly emerged as a crucial tool in several biological and chemical systems, including industrial, clinical, and fundamental research. It has enabled researchers, scientists, and engineers to perform numerous parallel experiments rapidly while consuming minimal reagents as opposed to simultaneously (and often sequentially) working with multiple conventional benchtop analytical instruments in a laboratory [1,2]. Microfluidic devices have an advantage over traditional in-vivo or in-vitro analysis systems by having more precise control over the physical parameters, sensor integration, and automation in a small form factor with a high degree of portability as well as energy and cost savings [3]. This has led to the emergence of the so-called “lab-on-chip” devices, making significant strides in diverse areas ranging from grand challenges such as water purification to fundamental research such as genetic analysis, organ on a chip, electrophoresis, molecular analysis, and pathogen detection [4,5].

Conventional microfluidic devices are largely fabricated through microfabrication techniques derived from the Micro-Electromechanical Systems (MEMS) and the semiconductor industry [1] involving photolithography, harmful chemical (photoresist), and expensive cleanroom facilities [6,7]. The process generally involves fabricating positive micron-sized extrusions on silicon wafers which then serves as a mold on which soft polymer (Polydimethylsiloxane (PDMS)) can be cured to encapsulate the wafer extrusions as hollow channels followed by bonding on a flat substrate (usually glass) to embed and seal all the microfluidic features. PDMS has become a staple in microfluidics chip fabrication for rapid prototyping due to its versatility, transparency, elasticity, biocompatibility, and reliability in replicating precise microfluidic features [8]. However, the process of developing molds from silicon wafers requires a cleanroom facility (class 10 or higher) [9,10], training to operate expensive and complex instruments such as the mask aligner [11], spin coater and high-power UV illuminators often requiring multiple days before the user can obtain a wafer mold. The final step in the assembly of the microfluidic chip requires plasma-assisted bonding of the cured PDMS on a glass slide followed by adding inlet and outlet ports with tubing connections [6,12]. The entire process is time-consuming, expensive, and requires a high level of technical expertise to complete. 

Alternate microfabrication techniques such as injection molding that enable rapid prototyping also exist [7,13] but are more suited for large-scale industrial production and lack the low start-up cost required for widespread use. Subtractive manufacturing processes, such as dry and wet etching [14,15], micro-milling [16], laser-cutting [17], xurography [18] and air-blasting [19] have demonstrated rapid microfabrication capabilities with different degrees of success however these techniques have certain limitations which currently outweighs the demonstrated advantages for microfluidic chip fabrication. Etching, xerography, and laser cutting produce curved, valley-like channels that are often imprecise for microfluidic applications [20,21]. Micro-milling offers rapid prototyping but generates channels with rough surface finishes and poses additional challenges at the chip bonding stage [16,22]. 

Industrial-grade and research stage 3D printing technologies such as selective laser sintering (SLS) [23] and Two-photon polymerization (TPP) [24,25] are currently being applied to print sub-micron sized features. Though these systems are revolutionary, the equipment cost is a major obstacle to their wide-scale application in prototyping microfluidic devices [26,27]. Recently, with the growing popularity, precision and affordability of desktop 3D printers, researchers have shown an increased interest in 3D printing with the aim of fabricating microfluidic chips in an inexpensive and simple manner to enable rapid prototyping [28,29,30]. However, the adoption of 3D printing in the microfluidic research community is slow due to the lack of resolution and versatility constraints imposed by these systems [31,32]. 

Most consumer-grade desktop printers work by splitting a 3D object model into several 2D layers (slicing) and materializing the object out of liquid polymer (melted thermoplastic or liquid resins) layer by layer. The two most common types of printing technology in this space rely either on fused deposition modeling (FDM, ~70% of market share) approach or stereolithography (SLA, ~15% of market share) approach [33]. FDM approach melts and extrudes thermoplastics locally to build a 3D object through a single printhead/extruder. Though there have been several works reported on fabricating microfluidic devices with transparent materials via FDM printing [34,35], most microfluidic systems fabricated from this technique lack micron-scale resolution and smooth surface finishes, are semi-transparent, suffer from common FDM print issues (wrapping, cracking, stringing) and are unable to yield well defined rectangular channel cross-sections [34,36]. SLA on the other hand uses a vat photo-polymerization process where the light of a specific wavelength is used to locally photopolymerize a pool of liquid resin layer by layer to yield a cured polymer solid object. Laser stereolithography [37], digital light processing (DLP) [38,39,40], masked stereolithography (MSLA), Material Jetting (MJ) [41], and continuous liquid interface projection (CLIP) [42] are some of the variations of this concept that deliver the same results but with different approaches of illuminating the resins. DLP-based printers, especially masked LCD printers (MSLA) have become ubiquitous in 3D printing with excellent community support. Due to the low barrier to entry, the adoption of MSLA technology in the microfluidic field including physical and biomedical applications has seen an upward trend in recent years [39,43,44].

One of the main components of SLA printing is the use of a photopolymerization resin [45]. The printing process takes advantage of the free radical polymerization that solidifies the liquid resin specifically on the point of incident light. Though the actual compositions for the commercial resins are kept undisclosed, the SLA resin generally consists of a monomer/oligomer for the building block and crosslinking, a photoinitiator initiating the radicalization process, a UV absorber to control UV penetration, and some pigment as an auxiliary agent for specific functionalities [46]. Nordin group did extensive work on optimizing the resin formulation to achieve good microfabrication results [47] and even developed an accompanying custom printer (with a custom LCD with 7.5 μm pixel size and light source) that could print microfluidic channels down to 18 × 20 μm resolution [43]. Mohamed et al. showed SLA printing can be used to print positive molds that can transfer the features via PDMS soft-lithography [48]. Zhang et al. and Ge et al. demonstrated printing hydrogel-polymer hybrid capable of having scaffold-like structures and absorbing capabilities using a custom printer-resin setup [49,50]. Folch group demonstrated 3D printed Quake valves [44]. However, these previous efforts either fabricated application focused microfluidic chips, use a custom-built SLA printer, or significantly modify existing SLA printers with custom parts to achieve these fine resolutions for microfluidic applications. While none of these are any major shortcomings of the prior research progress, there is still a gap that needs to be filled so that researchers can just start creating functional microfluidic devices with ready-made inexpensive MSLA printers. 

Here we provide application specific strategies that work with ready-made masked stereolithography (MSLA) 3D printing approach to fabricate functional microfluidic chips and microfluidic components. To bring additional functionality into fabricated microfluidic chips, we develop our custom UV curable resins incorporating the combination of high molecular weight Polyethylene glycol diacrylate (PEGDA) and Ethylene glycol polyether acrylate (EGPEA) as the monomer unit to print microfluidic components with tunable mechanical and optical properties. We analyze the range of achievable optical and mechanical characteristics as a function of varying monomer composition in our custom resins. Guidelines are provided to (i) create PDMS casted microfluidic chips generated from MSLA molds (requires plasma bonding), (ii) directly print microfluidic chips with embedded channels using commercial resins (no chip bonding steps required) and (iii) directly print microfluidic chips with embedded channels at a finer resolution and additional functionality using our custom resin formulation (no chip bonding steps). The user can choose either strategy based on their need and resource availability. Finally, we demonstrate further the utility of our custom resin by 3D printing an integrated pressure-actuated pneumatic valve (with flexible cured resin) in an otherwise rigid and clear microfluidic device that can be fabricated in a one-step process from a single CAD file. We also demonstrate the integration of a 3D printed finger actuated microfluidic pumps that can be easily interfaced with the inlet ports of microfluidic devices to move precise amounts of liquids within microfluidic devices without external syringe pumps.

## 2. Method and Materials 

### 2.1. MSLA Printing

We printed MSLA molds with positive extrusion (for PDMS casting) and embedded microfluidic channels using several consumer class desktop MSLA printers including Prusa SL1 (Prusa research, CR), Elegoo Mars 2 Mono and Elegoo Mars 3 Ultra 4k Mono (Elegoo, Shenzhen, China). The feature sets and cost of the printers are specified in Appendix A. All print geometries were designed with the opensource CAD software-FreeCAD (Version 0.19). The 3D.stl files were then sliced into several 2D sliced images using either PrusaSlicer (Prusa SL1), Sli3r (opensource), Chitubox (Elegoo Mars printers) enabling us to change the initial exposure time, exposure time, print orientation, support structures and layer heights. The Elegoo mars printer firmware also allows for print-pause-print operation enabling us to incorporate multiple resins in one a single print. 

### 2.2. PDMS Casting Using SLA Molds

The cured 3D parts were carefully taken off the build plate and were immediately submerged and washed with IPA to wash off any residual resin followed by re-rinsing with water to ensure a glossy finish. We found that this extra rinsing step helps in easy pealing of the PDMS from the SLA molds. The parts were then air dried and further cured under Prusa curing station (CW1S) for one minute. For PDMS casting, the PDMS (QSIL 216 Polydimethylsiloxane, a two-part elastomer encapsulant in 10:1 ratio) was poured over the SLA mold with positive features. The PDMS was cured in an oven (50 °C) for four hours. Post curing, inlet and outlet ports were cut into the PDMS followed by bonding it on a clean glass slide in a plasma cleaner (Harrick Plasma, PDC-32G, 115 V) for 30 s. The entire fabrication process-from concept to microfluidic chip-takes ~6 h (Figure 1A).

### 2.3. Droplet Microfluidic Chip

For droplet generation, we used mineral oil (standard oil, light; CAS: 8042-47-5) as the continuous phase, and water with blue food dye pigment as the dispersed phase. We used Span 80 (Sorbitan monooleate, Sorbitan oleate; CAS: 1338-43-8) as surfactant. Syringe pumps (New Era Programmable Pump Systems; NE-1000) were used to drive the dispersed and continuous phases at 10 and 100 μL/min respectively. The droplet generator chip and droplets were visualized using an optical microscope (Leica fluorescence microscope; 10–100×). 

### 2.4. Finger Actuated Screw Pump

The 3D printed finger actuated screw pump consists of two components-a screw and an inlet port (acting like a liquid reservoir)-that can be tuned to fit virtually any microfluidic device. The inlet port was designed to embed the negative threads to accommodate the screw. We used a standard size (M4–M6) screw thread since they provide several advantages over a custom thread design. The CAD model of the threaded screw can be found online and can be added to the chip design without any major modifications. To prevent liquid leakage due to the tolerances of the printed screws, we used silicone based vacuum grease (Dow Corning^®^ high vacuum grease) as a sealant and lubricant. Additionally, a rubber o-ring was added to the tip of the screw to provide an airtight contact between the screw and the microfluidic chip inlet port. The o-ring has an outer diameter corresponding to the diameter of the screw. We found that a syringe plunger gasket can also be used as an alternative to the o-ring for sealing the reservoir. 

### 2.5. Resin Formulation

For the commercial resins, we tried several off-the shelf inexpensive resins including Prusament tough resin (Prusa research, Prague, Czech Republic), Anycubic tough clear (Anycubic, Kowloon, China) and Nova3D tough clear resin (Nova3D, Shenzhen, China) that cure with 405 nm wavelength UV exposure. For our custom photopolymerization resin, we chose varying ratios of Ethylene glycol phenyl ether acrylate (EGPEA) (Sigma Aldrich, MO, USA, Cas: 48145-04-6) and Polyethylene glycol diacrylate (PEGDA) (avg M_n_: 700) (Sigma Aldrich, St. Louis, MO, USA; Cas: 26570-48-9) as the monomers. We chose Phenylbis (2,4,6-trimethylbenzoyl) phosphine oxide (Cas: 162881-26-7) (Also known as Irgacure 819) as the photoinitiator and 2-nitrophenyl phenyl sulfide (NPS) (Cas: 4171-83-9) as the UV absorber [43]. We prepared all variations of the custom resin at room temperature (23–25 °C). PEGDA-700 was stored in the refrigerator (2–8 °C) and was thawed at room temperature before being mixed with other reagents. The specified weight percentages were measured using a digital micro weighing scale, then the mixture was sonicated (DK Sonic Ultrasonic cleaner, 6 L, 180 W) for 30 min or until the mixture was homogenous. All concentrations of resin components are specified in *w*/*w* (weigh of the component/weight of the mixture) %. The prepared custom resins were stored at room temperature in dark jars to avoid curing from ambient light.

### 2.6. Directly Printing Microfluidic Chips with Embedded Channels

Directly printing microfluidic channels embedded in SLA prints eliminates the PDMS casting and chip bonding steps. We start with designing the 3D CAD model of the microfluidic chip and ensure the channels are within 1–4 slice layers deep from the chip surface. This avoids curing of the trapped liquid resin within the channels due to multiple UV exposure and thus significantly increases the probability of producing unclogged and clear channels (Appendix A). The CAD model is then sliced and printed using the recommended printer setting (Appendix A). Immediately after the print, the uncured liquid resin trapped within the microfluidic channels is flushed using compressed air through the inlet and outlet ports. The printed chips were then air dried and cured one final time in the Prusa curing station (CW1S) for one minute to yield the final functional microfluidic chip. The entire fabrication process-from concept to microfluidic chip-takes ~2 h (Figure 1B). To characterize the dimensions of microfluidic features (positive extrusions in SLA molds and internal channel cross-section of embedded channels), a brightfield optical microscope (Leica fluorescence microscope; 10–100× objective) was used to take high resolution images of the microfluidic features. An accompanying microscope image analysis software (ToupView) was used to quantify the length scales of the microfluidic features of interest. 

### 2.7. Integrated Pneumatic Valve

The CAD file for the multilayer microfluidic chip with liquid channels, flexible membrane and the rigid air chamber is provided as a Appendix A. Briefly, all channels have cross-section of 300 × 300 μm. The rigid material layers were printed with the commercial resin (Anycubic clear) and the flexible layers were printed with our custom flexible resin ([EGPEA:PEGDA] = [1:0.1] + 1% IRG + 1% NPS). The exposure times were set based on the optimal exposure time for respective resins (Appendix A). The print-pause-print method was invoked by accessing the firmware setting of the MSLA printer to enable multi-resin printing. The printed multilayer chip was then air dried and cured one final time in the Prusa curing station (CW1S) for one minute to yield the final functional microfluidic chip. The closing pressure of the pneumatic valve was measured by monitoring the pressure of the air chamber using an externally connected pressure transducer (4–10 mA output; Hilitand Cat #: Hilitand7py6g9uith-01) in line with the air supply line. Even though the one-step multi-material printing seems straightforward, there are certain steps the user should be aware of to consistently produce successful prints. For example, if the commercially available desktop printers don’t offer the ability to pause the print midway to swap resins, the user would either have to manually pause the print at a particular layer to swap the resins or change the printer firmware code to achieve the print-pause-print feature. Additionally, changing the resins in between prints also requires the user the change the initial exposure time to resin-specific settings. This would result in the user specifying at least 4 exposure times for a multi-material print job compared to specifying 2 exposure times (for initial and subsequent layers) for a single resin print job. The exposure time setting for each layer for the pneumatic valve chip is provided in Appendix A.

## 3. Results

### 3.1. Printing SLA Molds for PDMS Cast

First, we wanted to characterize the resolution of commercially available desktop MSLA printers and resin in fabricating positive extrusions to serve as SLA molds to generate PDMS casts. Even though the length of the positive extrusion can be oriented in either the horizontal x-y plane (parallel to the build plate) or along the z-axis (perpendicular to the build plate), we find that printing the extrusions on along the horizontal plane yields better results as the layer-by-layer print method ensures that features are properly aligned as they are simultaneously printed in the x-y plane. Whereas printing extrusions along the z-axis produces undesirable surface roughness and opacity (Appendix A). The theoretical print resolution in the MSLA printers is constrained by the pixel size of the masking LCD (also known as the x-y resolution) and the step size of the stepper motor mounted to the build plate (z resolution). We used several inexpensive and easily accessible desktop MSLA printers for our tests (Appendix A). Among these Elegoo Mars 3 has a 4k LCD screen yielding pixel size of 35 μm with a minimum vertical step size of 10 μm. The key parameters that dictate the efficiency of the curing process are the resin chemistry and its UV exposure characteristics. The UV intensity in typical desktop MSLA printers is fixed allowing the users to only tune the exposure time to optimize the print quality of microfluidic features. Exposure time primarily determines the extent of curing and thus the mechanical properties of the printed parts. The initial exposure time is set more than the print exposure time so that the first few layers can overcure such that a strong bond is made between these layers and the build plate to support the final print weight. For subsequent layers, under-exposure might not cure the resin at all while over-exposure of the resin results in thicker cured layers than intended and aggravates light bleed effect or elephant foot effect resulting in inaccurate print dimension [51]. Due to the smaller footprint of microfluidic devices, we found that an initial exposure time that is 4–5 times the normal exposure time yields good first layer adhesion throughout the print duration. We used a dimension calibration test file to optimize the printer exposure settings (Appendix A). 

To develop microfluidic PDMS casts, we designed SLA molds with several positive extrusions mimicking long straight channels with both 90° bends and smooth curved sections. PDMS was poured over the printed molds, cured, and bonded (plasma-assisted) to glass slides to yield sealed and functional microfluidic chips (Figure 1A). We measured the difference in dimension between the nominal dimension (obtained from the CAD models) and the printed channel dimensions for the positive extrusions to determine the error associated with the printing process. For channels with cross-section above 75 by 75 microns, the error is below ~5% and drops to below ~1% for channels with cross-section above 200 by 200 microns (Figure 2A). We were also able to print channels down the theoretical resolution of printer (35 by 35 microns) albeit with ~20% error in the cross-section dimensions. This higher error near the theoretical limit of the MSLA LCD screen can be attributed to two factors. First, the laser intensity across individual pixels in SLA-based printers follows a gaussian or parabolic profile [52,53]–translating into differential curing of resin at a single pixel level. Second, there is a vacuum created between the cured layer and fluorinated ethylene propylene (FEP) film (at the bottom of the resin vat) resulting in a momentary deformation in the FEP film after each curing and bed tilting step. Such FEP deformation results in deformed printed edges. Taken together, these effects yield positive extrusions with variable cross-sections at the sub-50-micron length scale. This technique enabled us to generate a PDMS cast droplet generator with a channel width of 100 microns yielding a stream of aqueous droplets (*D* = 100 μm) (Figure 2B).

We also demonstrate the use of MSLA printers with commercially available resins to print functional peripheral components that can be integrated with any existing microfluidic chip. For these tests, instead of using the SLA mold method, we directly printed microfluidic chips with hollow embedded channels (as described in Section 3.2). One of the major components of microfluidic systems is the need for external syringe pumps to drive the liquid through the microfluidic devices. The syringe pumps are often expensive, requires power and significantly increase the footprint and complexity of an otherwise miniaturized and portable system. We describe the fabrication and characterization of 3D printed finger-actuated displacement pump consisting of screw threads that seal and compress the fluid in a connecting reservoir (Figure 2C). The resulting setup enables on-chip fluid storage and its metered delivery into the microfluidic chip by twisting the screw cap by pre-determined angle of rotation with fingers (Figure 2D). The finger twist pumping mechanism is capable of injecting and withdrawing a precise quantity of liquid into the chip with different channel cross-sections using a pre-calibrated twist angle–injection volume plot (Figure 2E). Such 3D printed components are easy to fabricate and can be integrated with battery-operated programable motors to further enable portable flow control in microfluidic chips [54].

### 3.2. Direct SLA Printing of Microfluidic Chips with Embedded Hollow Channels

Directly printing embedded hollow microfluidic channels within SLA cured resins is a faster and a simpler method of developing microfluidic chips as it alleviates the need for PDMS casting and chip bonding. However, commercial SLA resins are not optimized to print microscopic hollow features often yielding clogged channels. The failure points at the micro scale can be attributed to two main effects (Figure 3A,B). First, printing embedded hollow channels results in liquid resin getting partially trapped inside the cavity of the liquid resins (Figure 3A). Uncured resins can be flushed out of the channels in the post-processing step resulting in unclogged functional channels. However, for channels with trapped liquid resin, the curing of subsequent layers below the hollow channel results in multiple partial exposures of the trapped liquid resin to additional curing cycles. Second, the deeper the channel is from the final surface of the chip, the more likely it is that the trapped liquid resin gets cured (due to several UV partial exposures cycles), resulting in channel clogging (Figure 3B). To address this, we formulated and evaluated several custom UV curable resins with tunable UV absorbance capacity. The UV curable resin typically consists of a monomer, a photoinitiator, a UV absorber and pigments for color or any additional characteristics [52,55,56]. For this study, we chose Polyethylene glycol diacrylate (PEGDA) (avg Mn: 250 & 700) and Ethylene glycol phenyl ether acrylate (EGPEA) as monomers, Phenylbis (2,4,6-trimethylbenzoyl) phosphine oxide (commercially known as Irgacure 819) as photoinitiator and 2-nitrophenyl phenyl sulfide (NPS) as the UV absorber based. NPS was selected to allow tunable UV absorption [47,56] and EGPEA/PEGDA monomers were selected to incorporate tunable mechanical properties of printed parts [43,56].

The interplay between light exposure and liquid resin curing is captured by the Beer-Lambert law [47,56].
(1)$$A=-\log_{10}\frac{P}{P_{0}}=\epsilon lc$$
where, *A* = Absorbance or optical density of the liquid resin; *P* = Intensity inside the resin; *P*_0_ = Incident light intensity; ϵ = absorptivity; l = length of the beam in the absorbing medium; *c* = concentration of the absorbing liquid resin. Assuming the absorbed energy required to cure the resin at the point of exposure is uniformly distributed along the curing layer, we can deduce the following relation between the *z*–the cured layer height and *t*–time since initial exposure,
(2)$$z=h_{d}\ln\left(\frac{t}{t_{0}}\right)$$
where $t_{0}$ is the duration of initial exposure and *h_d_* is the characteristic penetration depth of the resin. The characteristic penetration depth (*h_d_*) of different resins can be obtained experimentally by determining the slope of the cured layer height vs. exposure time curve. The photoinitiator (PI) and UV absorber play a key role in determining the radicalization efficiency and amount of absorbed light respectively. We evaluated the curing characteristics of our custom resins by changing the PI concentration from 0.1% (*w*/*w*) to 1% (*w*/*w*) and the UV absorber (NPS) concentration from 0.1% (*w*/*w*) to 1% (*w*/*w*). To determine and compare the *h_d_* of our custom resins and commercial resins, we printed a hollow channel with a cross-section of 300 × 300 μm and sealed it with a single layer on the top with a 25 μm layer height. In a series of experiments, the top layer was printed with increasing exposure times (*t*) from 1 to 15 s which resulted in different cured layer thicknesses (*z*). By plotting the measured cured layer thickness against the exposure time in a log-linear plot, we could quantify the characteristic penetration depth $\left(h_{d}\right)$ for individual resin from the slope of each line. Our analysis revealed that commercial clear resin (Anycubic) yielded an *h_d_* ~120 μm (Figure 3C). We find that at lower concentrations of NPS (0.5% (*w*/*w*)), the resin yields an *h_d_* ~150 μm. Both the commercial resin and custom resin with low NPS concentration yielded high penetration depths resulting in curing of trapped resin and clogging of the hollow embedded channels (Figure 3E). However, as we increase the NPS concentration (NPS-1% (*w*/*w*)), the *h_d_* value significantly decreases to ~60 μm providing the trapped liquid resins within the channels with more shielding against subsequent UV curing cycles and thus enabling easy removal of liquid resin in the post-processing steps, resulting in clear unclogged channels (Figure 3E). Additionally, working with resins with lower *h_d_* values also ensures that there will be a smaller variation in the cured layer height (10–100 μm for *h_d_* ~60 μm) as opposed to a larger variation in the commercial resins with high *h_d_* (150–400 for *h_d_* ~150 μm) under a similar range of exposure times (Figure 3C). We also verify that our custom resin formulation with high NPS concentration (1% NPS (*w*/*w*)) has significantly higher absorption capacity in the 350–400 nm than the commercial resin (Figure 3D). This can be attributed to its lower characteristic penetration depth. When we print multiple layers (in one-layer increments) to seal the embedded channel using commercial resin (Anycubic), we observe that after 4 subsequent layers, the trapped resins start to cure and clog the channel (Figure 3F). Whereas our custom resin with low penetration depth (*h_d_* ~60 μm) yields unclogged channel even after printing 6 subsequent bottom layers enclosing the channel (Figure 3F). The efficient absorption characteristic of resins with low penetration depth $\left(h_{d}\right)$ keeps the trapped resin uncured which can then be easily flushed out in the post-processing stage. 

To determine the reproducibility and confidence regimes for successfully printing embedded microchannels we conducted a z-test with the following null-hypothesis: The SLA resin can print microfluidic channels with unclogged embedded hollow channels with reproducibility of >90%. For each of our tested channel cross-section dimension ranging from 50–600 microns (in 50-micron increments), we printed 9 replicates using both the commercial (Anycubic clear) and our custom resin ([EGPEA:PEGDA] = [1:1] + IRG 819 (1% *w*/*w*) + NPS (1% *w*/*w*)), resulting in a total of 216 printed embedded channels. The channels were designed to contain both 90° bends and smooth curved sections with a total length of 1.5 mm. After washing the prints with IPA and drying, the microfluidic channels were tested by flushing water through the channels and visually inspecting the channel cross-section under a brightfield microscope to ensure that the channels were unclogged and had the correct cross-section dimensions (Figure 4A). While both resins could successfully print all channels over 350 × 350 μm, for the commercial resin the null hypothesis can be rejected for channels with cross-section more than 300 μm based on a *p* value of 0.00115 (criterion for rejection: *p* value < 0.01). Whereas for the custom resin, the null hypothesis can be rejected for channel cross-section more than 200 μm based on a *p* value of 10^−4^ (criterion for rejection: *p* value < 0.01). These statistical results suggest that under the optimal print setting, the commercial resin (Anycubic) can reliably and reproducibly print unclogged microfluidic channels with minimal channel cross-section of 300 microns. This limit can be further lowered to 200 microns when printing with a custom resin formulation (*h_d_* = 60 μm) with lower characteristic penetration depth (Figure 4B). Even though it is possible to print embedded channels with cross-section as low as 150 × 150 microns and 100 × 100 microns using the custom resin, the frequency of printing unclogged channels drops at these smaller length scales. We also find that for smaller channels (i.e., channels with nominal width less than 350 microns), the printed cross-section of channel deviates from the square shape. This observation is somewhat alleviated when using our custom resin formulation (1% NPS) but even here, we observe non-square channels below a nominal channel width of 250 microns (Figure 4A). This discrepancy in the printed shape of the cross-section can be attributed to the dominance of surface tension driven interfacial forces between the cured and un-cured resin over gravitational force at this length scale that forces the resin to form a meniscus around the sharp corners. Additionally, effects like UV light bleed and refraction due to the hollow air pocket in the channel may further contribute to the distortion of the final printed shape of the hollow channels.

### 3.3. Multi-Material SLA Printing with Tunable Mechanical Properties

Developing a custom resin formulation also enables us to vary the ratio of the monomers to formulate resins that cure to yield different mechanical properties. While several studies have been conducted to determine the mechanical properties of single monomer resin [49,56,57,58,59], the characterization of mechanical properties of a cured resin composed of multiple monomers hasn’t been studied extensively. We choose a mixture of two acrylates–EGPEA and PEGDA-700-as monomers at different concentrations in our custom resin formulation. The main factors contributing to the flexibility of polymer chains are the number of cross-links, type of side chains and freedom of rotation along C-C bonds [60,61]. EPGEA is a mono-acrylate with a stable phenyl group at one of its ends which does not participate in the cross-linking process. This reduces the number of cross-links resulting in shorter polymeric units that yield flexible cured resin. On the contrary, the di-acrylate PEGDA has an acrylate group on both sides that contribute to the cross-linking process. This generates more cross-linking during the polymerization process resulting in long chained polymeric units that yield hard and brittle cured resin. To characterize the chemical properties of the resin mixture and individual monomeric constituents, we performed Fourier-transform infrared spectroscopy (FTIR). Since both monomers have acrylate groups with a long carbon chain, the differences in peaks can be found in the fingerprint region of the spectra (600–1400 cm^−1^). EGPEA monomer shows characteristic peaks at the 750–690 cm^−1^ and 1490 cm^−1^ regions corresponding C=C bending and C=C stretching frequencies respectively. PEGDA-700 monomer shows a characteristic broad peak at 1111 cm^−1^ region corresponding to ester linkage. First, the mixture with equal amounts of the two monomers (i.e., [EGPEA:PEGDA] = [1:1]) retains the characteristic peaks of both EGPEA and PEGDA monomeric resins (Figure 5A). Second, at low concentrations of PEGDA ([EGPEA:PEGDA] = [1:0.1]), the resin mixture predominantly exhibits EGPEA cross-linking as evident from the disappearance of the ester linkage peak. This results in more flexible cured prints as characteristic of EGPEA cured resins. As the PEGDA composition increases, the peak associated with the ester linkage reappears signifying contributions of both polymeric units in the final resin mixture. High PEGDA concentration ([EGPEA:PEGDA] = [1:2]) yields hard and brittle cured prints due to the dominance of bifunctional diacrylate (Figure 5A).

To characterize the mechanical properties of the custom resins, we performed tensile stress tests on the printed parts comprising of varying concentrations of PEGDA ranging from [EGPEA:PEGDA] = [1:0.1]–[1:2]. We changed the monomer ratio based on the weight percentage and kept the photoinitiator as well as the UV blocker concentration constant (1% IRG 819 + 1% NPS) for all variations. The resins were printed to form tensile specimens resembling a dog-bone shape to ensure that most of the tensile stress is concentrated in the middle of test specimen (i.e., in the gauge section) forcing the specimen to break at that region. The setup enabled us to obtain standardized stress-strain profiles for printed parts for a range of resin compositions (Figure 5B). From these, stress-strain profiles we were able to compare the Young’s modulus of our custom resins. Resins with low concentrations of PEGDA monomer yielded printed parts that are flexible with low Young’s modulus. As the PEGDA concentration is increased relative to the EPGEA, the Young’s modulus of the printed part increases resulting in less flexible and rigid printed parts (Figure 5C). The details of our custom stress testing setup are provided in the Appendix A.

### 3.4. One-Step Multi-Material Printing of Chips with Integrated Pneumatic Microfluidic Valve

By using two different resins in a single print, we were able to incorporate resins with both high and low Young’s moduli in one microfluidic chip. The firmware of the commercially available MSLA printers can be programmed to automatically switch resins during a single print using print-pause-print method enabling us to print such hybrid microfluidic chips. This enabled us to print an entire microfluidic chip that comprised both flexible and rigid parts in spatially separate layers. To test the functionality of this concept, we designed a multi-layer microfluidic chip consisting of a flexible membrane that acts as a pneumatic valve sandwiched between a top layer of a rigid microfluidic flow channel and a bottom layer of rigid compressed air chamber (Figure 6A,D). The flexible valve function by deforming under the action of the closing pressure (Δ*P_c_* = *P_l_* − *P_a_*) which is the pressure differential arising from the difference in pressure of the flowing liquid (*P_l_*) in the microfluidic channel and applied pneumatic pressure (*P_a_*). The closing pressure then becomes the minimum pressure required to completely close the pneumatic valve and halt the liquid flow in the microfluidic channel (Figure 6B). Several pneumatic valves can be integrated in a single microfluidic channel that can be independently actuated through separate compressed air lines operated either manually (via hand driven syringe) or programmatically (via automated pressure control lines) to enable complex flow control in microfluidic chips (Figure 6C). We characterized these valves by driving water (with blue dye for visualization) through the microfluidic flow channel with a constant inlet pressure source. A secondary syringe with air was used to pressurize the air chamber beneath the flow channel. The compressed air chamber pressure and liquid flow rate were simultaneously monitored in real-time to determine the closing pressure corresponding to the no flow state. The response of our pneumatic valve to external air pressure ultimately depends on the thickness of the valve membrane and Young’s modulus of resin formulation. For example, for a fixed resin formulation with low concentration of PEGDA ([EGPEA:PEGDA] = [1:0.1]), the closing pressure monotonically increases as the membrane thickness is increased (Figure 6E). Alternatively, for a fixed membrane thickness of 150 μm, the closing pressure monotonically increases as the concentration of PEGDA in the resin formulation increases (Figure 6F). Taken together, the valve closing pressure can be tuned independently by selecting valve thickness and the concentration of PEDGA with respect to EGPEA in the resin formulation. Additionally, we verified the biocompatibility of the 3D printed cured resins by successfully conducting nucleic acid amplification via Loop mediated isothermal amplification reaction (LAMP) in 3D printed chips. The 3D printed chips were used to house and incubate the amplification assay at 65 °C for 30 min, amplifying a target-specific sequence of λ DNA (Appendix A).

## 4. Conclusions

3D printing technology has come a long way since its inception and continues to break accessibility barriers between consumers and research pursuits. The open hardware movement in the consumer 3D printing space is continuously improving the print resolution and accuracy while simultaneously reducing the cost of these instruments. Prototyping microfluidic devices for research and development has yet to fully harness the availability of off-the shelf inexpensive desktop MSLA 3D printers. We address this gap by providing guidelines to researchers and the science hobbyist community to rapidly fabricate microfluidic devices and choose the appropriate microfabrication strategy based on application specific needs. These techniques work completely with existing consumer class MSLA printers (ranging from ~$200–$2000) without any modifications to accurately print (i) PDMS casted microfluidic channels with channel sizes as low as ~75 μm and (ii) embedded channels with channel size as low ~300 μm (~200 μm with custom resin formulation). We anticipate these resolutions will only improve in the coming years as the pixel density (ppi) of the new LCD displays continues to increase following Moore’s law trend. Additionally, such printers can also be used to fabricate integrated microfluidic components such as finger-actuated microfluidic pumps and flexible pneumatic microfluidic flow valves. The convenience and swiftness of our proposed technique enable one to go from an “idea stage” to a “functional microfluidic chip” within ~hours as opposed to traditional clean room-based microfabrication (~days), which could drastically lower the barrier to entry in microfabrication and greatly accelerate lab-on-a-chip research.

## Figures and Tables

**Figure 1 biosensors-12-00652-f001:**
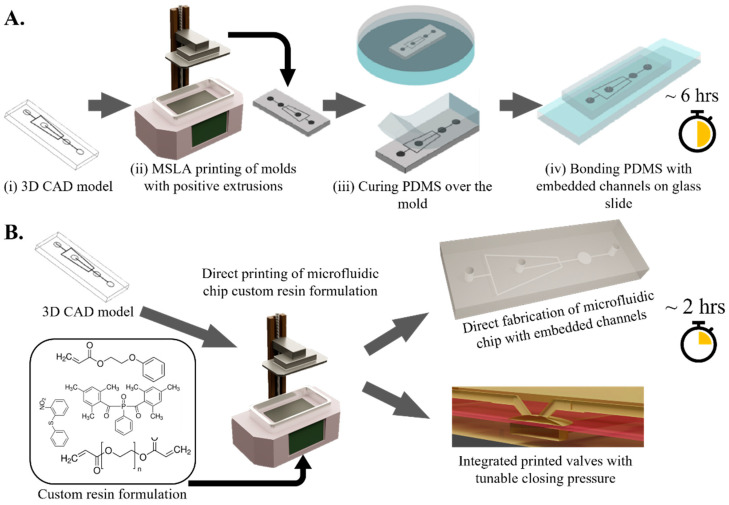
(**A**). PDMS microfluidic chips developed with 3D printed positive molds. Fabrication steps include (i) developing a 3D CAD model of the positive mold, (ii) stereo-lithographically printing and curing the positive molds, (iii) casting and curing PDMS over the positive molds and (iv) plasma assisted bonding of the cured PDMS with embedded channels onto a glass slide. (**B**). Incorporation of our custom resin formulation with existing desktop MSLA printers to enable direct fabrication of microfluidic chips with embedded channels without PDMS casting and curing. Our custom resins also enable us to simultaneously incorporate multi-resin composition with tunable mechanical and optical properties in a single chip.

**Figure 2 biosensors-12-00652-f002:**
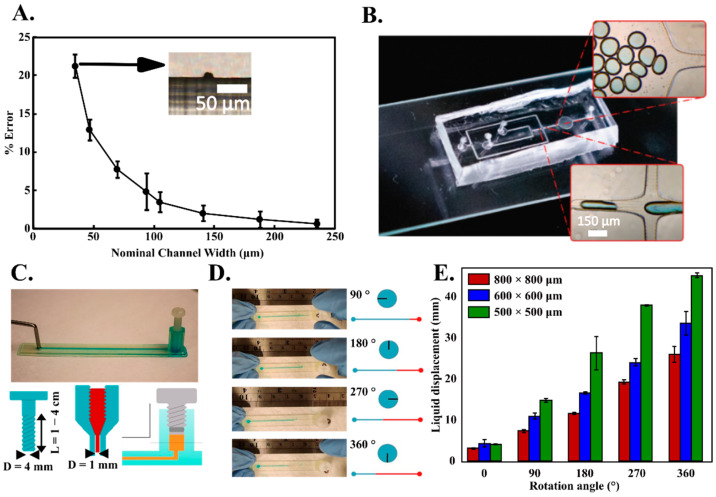
(**A**). Error between the nominal and printed channel dimensions using the SLA molds for PDMS casts to develop microfluidic devices. (**B**). A microfluidic droplet generator developed with an SLA printed mold yields channels with a width of 100 μm capable to generating aqueous droplets (**C**). 3D printed finger twist screw with a corresponding fluid reservoir can be integrated with microfluidic channels to enable pump free fluid transport. (**D**). Finger-actuated fluid pumping enables precise liquid metering in a microfluidic chiap. (**E**). Tunable and precise liquid injection into microfluidic chips based on the screw rotation angle in different sized microfluidic channels.

**Figure 3 biosensors-12-00652-f003:**
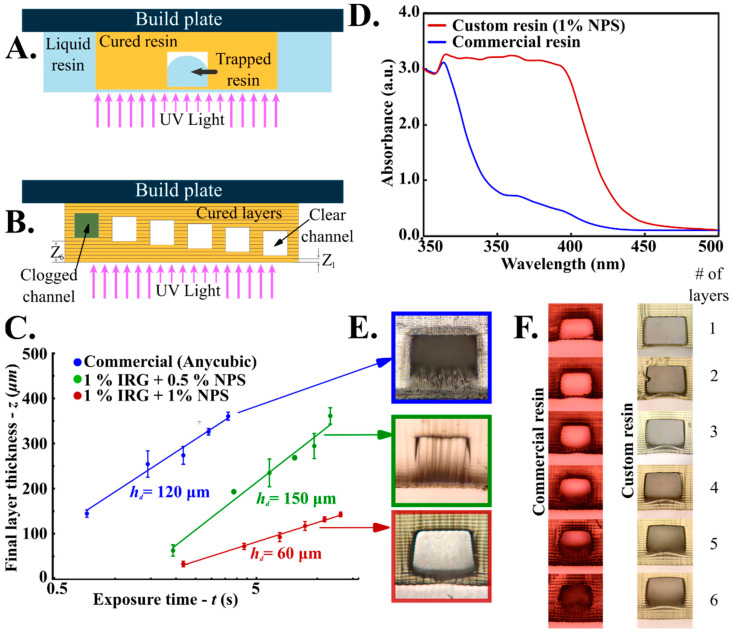
Schematic illustrating (**A**). trapped liquid resin in an embedded channel and (**B**). curing of the trapped resin due to multiple UV exposure cycles. (**C**). Log-linear plot for exposure time (*t*) vs. final layer thickness (*z*) for commercially available and custom resin formulations. (**D**). UV/Vis spectra of commercial and custom resin formulation with 1% NPS. (**E**). Microscopic images of 300 micron square channel cross-section printed with commercial resin with *h_d_* = 120 μm (blue box, partially clogged), custom resin with low UV blocker composition with *h_d_* = 150 μm (green box, fully clogged) and custom resin with high UV blocker composition with *h_d_* = 60 μm (red box, clear). (**F**) Intermediate images of the microfluidic channel cross-section (for 300 μm square channel) after curing of each successive layer to seal the channel using both commercial and custom resins.

**Figure 4 biosensors-12-00652-f004:**
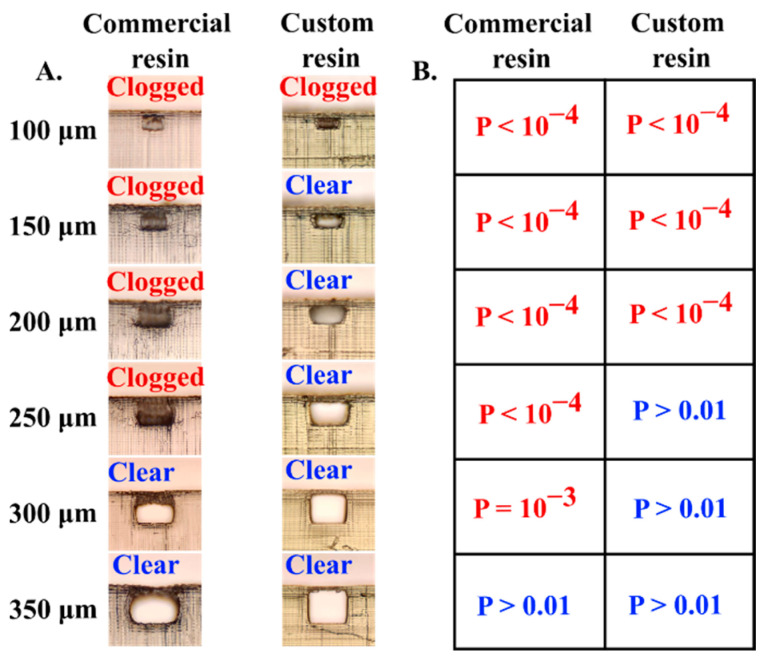
(**A**). Cross-sectional profile of embedded square channels) printed from commercial and custom resin (1% NPS) formulation to characterize successful and unsuccessful prints. 9 replicates were performed for each channel cross-section ranging from 50–600 μm. (**B**). Corresponding *p* values from a z-test (single tail) were calculated to determine the statistically reliable probability of printing clear unclogged channels [Null-hypothesis: The SLA resin can print microfluidic channels with unclogged embedded hollow channels with reproducibility of >90%; Criterion for rejection of; *p* value < 0.01].

**Figure 5 biosensors-12-00652-f005:**
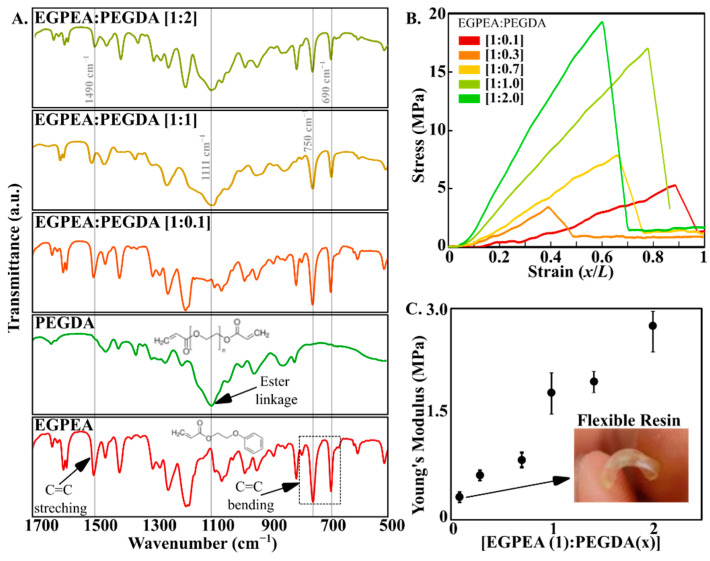
(**A**). FTIR spectra of custom resin formulations with varying amounts of EGPEA and PEGDA. (**B**). Stress-strain profiles, obtained from tensile stress tests performed on custom resin prints. (**C**). Young’s modulus of custom resin prints as function of varying PEGDA concentration.

**Figure 6 biosensors-12-00652-f006:**
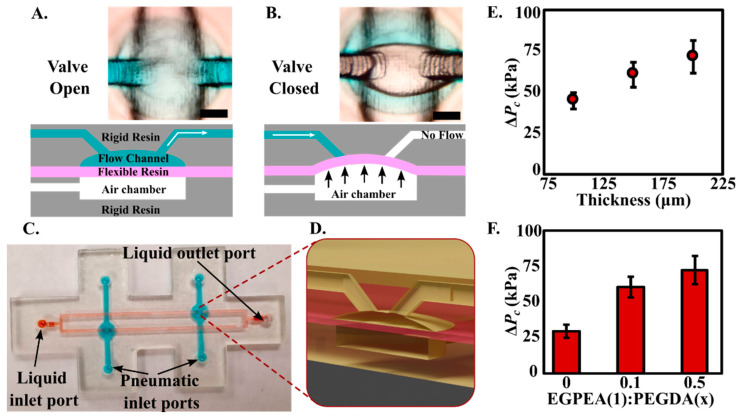
Top view (chip valve image) and side view (illustration) of the flexible pneumatic valve sandwiched between rigid flow channel and rigid compressed air chamber in the (**A**). valve open and (**B**). valve close sates (scale bar = 300 μm). (**C**). 3D printed Microfluidic chip incorporating separate inlet port, outlet port, channels for the liquid flow, and compressed air lines. (**D**). 3D blowout of the pneumatic valve illustrating the location of the flexible membrane (in red). Variation of valve closing pressure as a function of (**E**). membrane thickness and (**F**). resin composition.

## Data Availability

Not Applicable.

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
