# Peer review of "Multi-Resin Masked Stereolithography (MSLA) 3D Printing for Rapid and Inexpensive Prototyping of Microfluidic Chips with Integrated Functional Components"

_biosensors, 2022, doi:10.3390/bios12080652_

Round 1
Reviewer 1 Report
The manuscript by Ahmed et al. presents evidence that 3-D microfluidics can be made rapidly by commercially available stereolithography systems. By incorporating two resins, one firm and one soft, they demonstrate the ability to create Quake valves. The work is interesting, clearly written, and their data support all of their claims. The field of microfluidics, it could be argued, is moving in this direction of single step manufacturing compared to its origins in soft lithography, however, I think this contribution is only a minor advancement. Nevertheless, I am inclined to think it will be of interest to the readers of this special issue of Biosensors about 3D printing. I recommend it with the following suggestions/comments:
1. For Fig 2A The Channel Cross Section should be relabeled nominal channel width, since cross section implies an area. I believe the %error is only for the channel width and not the height, but if the height data is available maybe it could be added to a second axis or another graph.
2. In the text, the twist angle injection volume plot (Fig 3E) should be (Fig 2E).
3. In line 325 the authors suggest two failure points for making embedded channels by SLA and parenthetically refer to Fig 3A and 3B, but I think it would be helpful to the reader to state the two points in words as well in that paragraph.
4. Fig 3D is odd. In the caption it is labeled as a UV/vis spectra, which would imply either an absorbance scale or a percent transmittance scale, but the scale reads “intensity”. More importantly the blank (water) should be a flat line since water does not absorb at any of those wavelengths. The resin should absorb a lot in the UV (350-400 nm) and less so in the visible, which makes me think it is a %T scale, but in any event it is unclear.
5. In line 387 “adsorption” was used instead of absorption
6. I think it was implied that for the embedded channels in Figure 4, the channel cross-sections were supposed to be square. If not the authors should clarify, and the authors should acknowledge that the cross-sections are not square and explain briefly why not. (unless they were supposed to be rectangular, in which case that should be stated)
7. Figure 6A,B have scale bars but their size is not indicated in the caption.
8. I think some comment on the clarity of the devices seems relevant if these are to be used for biological applications. Can you see cells inside one of these acrylic devices? It seems from Fig 6 that they have some clarity but a statement about it would be helpful.
9. Acrylates and the photoinitiator may leach out of the device over time. Again if these are to be used in biological assays, it would be helpful if the authors could comment on their biocompatibility.
Author Response
Reviewer response sheet is attached.

Reviewer 2 Report
This is an interesting paper showing a simpler and faster method/technique to implement microfluidic chips. The English style is good and the manuscript language is clear.
The introduction section presents a contextualization of the issue, a very complete state of the art, and clarifies the main objectives of the work developed. Some issues must be improved:
- Line 115, 116: should be Mohamed et al. / Zhang et al. and Ge et al.
- Line 106: it is missing a space between the word and the reference: resin[45].
The methods section describes the materials and methodologies used for the experimental work. Figure 1A should be labeled with (i), (ii), (iii), and (iv), as refer on the text.
The results section is well organized and the presented results are extensively discussed. However, I have some concerns:
- Figure 2E presents the liquid displacement as a function of the rotation angle, for different channel cross-sections. The lower is 500×500 µm2. I was expecting to present these results for lower values for the channel cross-sections since previously, it was concluded that this method can achieve approximately 200×200 µm2 (or a bit lower) with low error dimensions.
- Line 305: Please correct to Fig. 2C, instead of Fig 2C.
- Line 310: Please correct to Fig. 2E, instead of 3E.
- In Figure 3D should be clear that transmittance curves are presented for different resins, according to the discussion of the results. I suggest correcting the yy axis to “transmittance intensity (a. u.)”.
- Are Figures 3C, 3D, and 3F for 200×200 micron square channel cross-section (as referred at the figure label) or for channels with a cross-section of 300 μm x 300 μm (line 369)?
- Section 3.4 is a bit different from other results sections since it refers to multi-layer printing. What are the main challenges of this approach compared with single-material printing (with commercial or customized resin)?
Overall this new method can for sure revolutionize lab-on-a-chip research. Still, some points must be carefully addressed such as the comparison between this new technique and the conventional clean room microfabrication methods. The latter ones allow us to achieve lower resolutions despite the higher fabrication costs.
Author Response
Reviewer response sheet is attached.

Round 2
Reviewer 2 Report
The authors replied to all my comments and have changed the manuscript accordingly.